# Performance of Thermal-Oxidative Aging on the Structure and Properties of Ethylene Propylene Diene Monomer (EPDM) Vulcanizates

**DOI:** 10.3390/polym15102329

**Published:** 2023-05-16

**Authors:** Quanchao Hu, Qiang Chen, Peiru Song, Xingyu Gong, Junyi Chen, Yongxian Zhao

**Affiliations:** 1Key Laboratory of Rubber-Plastics, Ministry of Education, Shandong Provincial Key Laboratory of Rubber-Plastics, School of Polymer Science and Engineering, Qingdao University of Science and Technology, Qingdao 266042, China; 2Keshun Waterproof Technology Co., Ltd., Foshan 528303, China

**Keywords:** EPDM vulcanizates, thermal-oxidative aging, thermal decomposition kinetics, antioxidant

## Abstract

A thermal-oxidative aging test at 120 °C was conducted on ethylene propylene diene monomer (EPDM) vulcanizates of the semi-efficient vulcanization system. The effect of thermal-oxidative aging on EPDM vulcanizates was systematically studied by curing kinetics, aging coefficient, crosslinking density, macroscopic physical properties, contact angle, Fourier Transform Infrared Spectrometer (FTIR), Thermogravimetric analysis (TGA) and thermal decomposition kinetics. The results show that the content of hydroxyl and carbonyl groups as well as the carbonyl index increased with increasing aging time, indicating that EPDM vulcanizates were gradually oxidized and degraded. As a result, the EPDM vulcanized rubber chains were crosslinked with limited conformational transformation and weakened flexibility. The thermogravimetric analysis demonstrates that the thermal degradation of EPDM vulcanizates had competitive reactions of crosslinking and degradation, and the thermal decomposition curve can be divided into three stages; meanwhile, the thermal stability of EPDM vulcanizates gradually decreased with increasing aging time. The introduction of antioxidants in the system can promote the crosslinking speed and reduce the crosslinking density of EPDM vulcanizates while inhibiting the surface thermal and oxygen aging reaction. This was attributed to the fact that the antioxidant can reduce the thermal degradation reaction level, but it is not conducive to the formation of a perfect crosslinking network structure and reduces the activation energy of thermal degradation of the main chain.

## 1. Introduction

Ethylene propylene diene copolymer rubber (EPDM) not only shows excellent flexion resistance, resilience, and low-temperature resistance but also has suitable chemical structure stability. It is widely used in the automotive industry, wire and cable, sealing materials and waterproof coiled materials, and other fields. Although EPDM possesses excellent stability in chemical structure, it will still age to a certain extent under the action of external factors such as light, oxygen, heat, and chemical media in the actual service and life process, which will degrade its performance and shorten the service life of products. Therefore, it is important to study the aging law of EPDM.

The degradation process of EPDM is a complex and multi-stage mechanism that can be separated into several different steps [1]. It can lead to changes in material properties, which are caused by physical and irreversible chemical processes, such as polymer chain breaking and crosslinking [2,3]. These processes substantially affect the technically relevant properties of polymer materials, such as strength, stiffness, toughness, and modulus [4]. Generally, the extent of aging-related changes is dependent on temperature and other aging conditions [5]. For example, the crosslinking density depends mainly on the material-dependent oxygen absorption capacity, rate of diffusion, and oxygen permeability [6], as well as the vulcanization temperature, pressure, and time. The aging of vulcanized rubber was shown to be related to a conversion of poly-sulfidic sulfur bridges to mono- and di-sulfidic sulfur bridges [7].

EPDM rubber has been studied in thermal oxygen aging [8,9,10,11], photo-oxygen aging [12,13], artificial weathering aging [14], radiation aging [15,16], and medium aging [17,18]. Its research methods and characterization methods are diverse, such as dynamic mechanical property analysis, spectral analysis (Fourier infrared spectroscopy, Raman spectroscopy), energy spectrum analysis (X-ray electron spectroscopy, nuclear magnetic resonance), and thermal analysis (differential scanning calorimetry, thermogravimetric analysis), etc. However, most of them are limited to the research of EPM and EPDM rubber. The research on the thermal oxygen aging of EPDM vulcanizates with antioxidant content and semi-efficient vulcanization system is few.

The most common aging type was thermal oxygen, which has a great effect on the performance of EPDM rubber. Therefore, the antioxidants that could delay aging and prolong the service life of rubber products by inhibiting oxidation and heat play a vital role in maintaining the property of rubber. The commonly used antioxidants include N-isopropyl-N′-phenyl-p-phenylene diamine (4010-NA) [19,20], 2-sulfur-benzimidazole (MB) [21], N-4(phenylephenyl)-maleimide (MC) [22] and 2,2,4-thrimethyl-1,2-dihydroquinoline polymer (RD) [23,24]. Here, EPDM vulcanized rubber with a semi-effective vulcanization system was subjected to thermal and oxygen aging tests at 120 °C (Figure 1). The effects of thermal oxygen aging on the structure and properties of EPDM vulcanizates were studied in detail through physical properties, aging coefficient, surface properties, crosslinking density, microstructure, thermodynamic property, and vulcanization characteristics of EPDM vulcanizates.

## 2. Experimental

### 2.1. Materials

EPDM rubber (3760P) was purchased from Dow Chemical of the United States; carbon black (N330) was purchased from Cabot Corporation; naphthenic oil (KN4006) was purchased from China National Petroleum Corporation. Sulfur, zinc oxide (ZnO), stearic acid (SA), tetramethylthiuram disulfide (TMTD), Poly(1,2-dihydro-2,2,4-trimethylquinoline) (RD), and N-Isopropyl-N′-phenyl-p-phenylenediamine (4010-NA) were all commercial industrial products.

### 2.2. Method

#### 2.2.1. Preparation of EPDM Vulcanizates

A two-stage mixing process was applied to EPDM vulcanizates by a Haake torque rheometer (Thermo Fisher Scientific, Poly Lab OS, Waltham, MA, USA) in this experiment. The speed of the internal mixer was set at 60 r/min, and the initial temperature was 90 °C. EPDM, ZnO, SA, carbon black (N330), and naphthenic oil were added in sequence. After dumping and pouring into the mixing mill, the two-stage mixing process was carried out. Later, sulfur and accelerator were added while the second temperature was 50 °C. After storage for more than 16 hours, the EPDM compound was vulcanized to the required samples with a plate vulcanizer (Huzhou Dong-fang Rubber Machinery Co., Ltd., XLB-400 × 400, Huzhou, China) under the pressure of 15–20 MPa and temperature of 170 °C. The formulation of EPDM vulcanizates is listed in Table 1.

#### 2.2.2. Thermal-Oxidative Aging

According to ISO 188:2011 standard, the thermal-oxidative aging test was carried out on samples in the thermal oxygen aging test chamber (GOTECH TESTING MACHINES Inc., GT-7017-E, Taipei, Taiwan, China). The settings of the apparatus were as follows: temperature of 120 °C, temperature deviation of ±2 °C, air flow rate of 0.5–1.5 m/s, and air exchange rate of 3–10 times/h. The aging times were 0 h, 24 h, 72 h, 120 h, 168 h, and 336 h, respectively.

### 2.3. Measurements

#### 2.3.1. Vulcanization Characteristics

The vulcanization characteristics are tested using a vulcanizer (GOTECH TESTING MACHINES Inc., GT-M2000-A, Taipei, Taiwan, China) according to GB/T 16584–1996 with a specimen weight of approximately 5 g in 170 °C for 30 min.

#### 2.3.2. The Crosslinking Density

The rubber only swells and does not dissolve in the solvent, the mixing entropy increases and the swelling equilibrium is reached when the Gibbs free energy Δ*G_s_* is zero (Formula (1)).
(1)$$∆G_{s}=∆G_{m}+∆G_{el}=0$$
where Δ*G_m_*, Δ*G_el_* are mixing free energy and elastic free energy change, respectively. In this experiment, the equilibrium swelling method was used to determine the crosslink density of EPDM vulcanized rubber. Assuming uniform deformation of the rubber in the solvent, the Flory–Rehner equation yields Equation (2). Cyclohexane was chosen as the solvent to determine the mass and density of EPDM before swelling, then put into the cyclohexane solvent, fully swelling, and test the mass after swelling every 24 h until the difference between the two measurements was less than 0.01 g, and calculate the crosslink density according to the following Equation (2).
(2)$$\upsilon_{e}=-\frac{1}{V}\left[\frac{\ln\left(1-v_{2}\right)+v_{2}+{Xv_{2}}^{2}}{{v_{2}}^{\frac{1}{3}}}\right]$$
(3)$$v_{2}=\frac{v_{1}}{v_{1}+v_{3}}$$
(4)$$v_{3}=\frac{m_{2}-m_{1}}{\rho_{3}}$$
(5)$$v_{1}=\frac{m_{3}}{\rho_{1}}$$
where *v_e_* is the crosslink density (mol/kg); *V* is the molar volume of solvent (L/mol); *v*_1_ is the volume of the rubber phase (m^3^); *v*_2_ is the volume fraction of the rubber phase in vulcanizate (%); *v*_3_ is the volume of solvent in the rubber phase after swelling (m^3^); *X_cyclohexane_* is the interaction parameter between rubber and solvent: (*X_cyclohexane_*: 0.35144524); *ρ* is the density of rubber (g/cm^3^); *ρ*_1_ is the density of reagent for swelling (g/cm^3^); *m*_1_ is the mass of vulcanized rubber before swelling (g); *m*_2_ is the mass of vulcanized rubber after swelling (g); *m*_3_ is the mass of the rubber phase (g).

#### 2.3.3. Physical Property

Tensile properties and hardness are measured using an AI-7000S electronic tension machine (GOTECH TESTING MACHINES Inc., Taipei, Taiwan, China) and shore A hardness tester (GOTECH TESTING MACHINES Inc., Taipei, Taiwan, China) according to ISO 37: 2005 and ISO 7619-2: 2004, respectively. The rate of change of performance is calculated according to the following Equation (6).
(6)$$P\left(\%\right)=\frac{X_{a}-X_{0}}{X_{0}}\times 100$$
where *P* is the rate of change of performance, %; *X_a_* is the measured value of the performance of the specimen after aging; *X*_0_ is the measured value of the performance of the specimen before aging.

#### 2.3.4. Water Contact Angle Test

The water contact angle is measured with contact angle meter (Shanghai Zhongchen Digital Technology Equipment Co., Ltd., JC2000D2, Shanghai, China). The measured liquid is distilled water, and the test temperature and humidity are 15 °C and 50%, respectively.

#### 2.3.5. Aging Coefficient

The aging coefficient (AC) [25] is the standard for evaluating the thermal aging performance of rubber and can be expressed as:(7)$$AC=\frac{\sigma_{2}\times \varepsilon_{2}}{\sigma_{1}\times \varepsilon_{1}}$$

In Equation (7), AC refers to the aging coefficient of rubber; *σ*_1_ and *σ*_2_ refer to the tensile strength (MPa) before and after aging; *ε*_1_ and *ε*_2_ represent the elongation at break (%) before and after aging, respectively.

#### 2.3.6. Attenuated Total Reflection Fourier Transform Infrared Spectrometer (ATR-FTIR) Analysis

The FTIR spectra of samples were obtained by using a Nicolet IS 10 FTIR spectrometer (Thermo Fisher Scientific, Waltham, MA, USA) with an ATR accessory. The measurements were made with a signal average of 10 co-added scans with the resolution of 4 cm^−1^ and a range of 600~4000 cm^−1^.

#### 2.3.7. Thermogravimetric Analysis

The thermogravimetric test was carried out in a N_2_ atmosphere. The samples were heated from 30 °C to 600 °C at the rate of 20 °C/min, and the samples were kept at 600 °C for 10 min. Then, the atmosphere was switched to air with the same rate and a 10 min constant temperature after heating to 800 °C.

#### 2.3.8. Thermal Degradation Kinetics

The thermogravimetric analysis plays an important role in the structure and thermal stability of polymers. The relationship between quality and temperature or time is tested under a certain flow rate atmosphere, so as to analyze the thermal stability and thermal decomposition of samples.

The degradation reaction equation is usually used to describe its kinetics.
(8)$$\frac{d_{\alpha }}{d_{t}}=kf(\alpha )$$
(9)$$G\left(\alpha \right)=kt$$
where α refers to the conversion rate, *k* refers to the rate constant, and *f*(*α*) depends on the reaction mechanism. The relationship between the rate constant *k* and the reaction temperature *T* can be expressed by the Arrhenius equation:(10)$$k=Aexp(\frac{-E}{RT})$$
where *A* is pre-factor (min^−1^); *β* is heating rate (°C/min); *T* is temperature (K); *E* is the activation energy (kJ/mol); *n* is reaction order; and *R* is molar gas constant (8.314472 J/(mol.k)). For the simple reaction equation *f*(*α*).
(11)$$f\left(\alpha \right)={\left(1-\alpha \right)}^{n}$$

Equation (12) can be obtained by substituting Equations (10) and (11) into Equation (8):(12)$$\frac{d_{\alpha }}{d_{t}}=Ae^{\left(\frac{-E}{RT}\right)}{\left(1-\alpha \right)}^{n}$$

The Freeman–Carroll method [26] is a method to calculate the activation energy by obtaining the difference between two points according to the reciprocal of the weight-loss rate and temperature at several points on a thermogravimetric curve (TG curve). Assuming the heating rate at a constant $\beta =\frac{d_{T}}{d_{t}}$, Equation (13) can be obtained from the dynamic equation and Arrhenius equation:(13)$$\frac{d_{\alpha }}{d_{T}}=\frac{A}{\beta }e^{\left(\frac{-E}{RT}\right)}{\left(1-\alpha \right)}^{n}$$

By taking logarithms on both sides of Equation (13) and dividing them by $∆\mathrm{lg}\left(1-\alpha \right)$, Equation (14) can be obtained:(14)$$\frac{∆\mathrm{lg}\left(\frac{d_{\alpha }}{d_{T}}\right)}{∆\mathrm{lg}\left(1-\alpha \right)}=-\frac{E}{2.303R}\times \frac{∆\left(\frac{1}{T}\right)}{∆\mathrm{lg}\left(1-\alpha \right)}+n$$

Adopting a temperature range of 450–480 °C and a temperature interval of 2 °C, this paper performed the Freeman–Carroll operation. The activation energy *E* and the reaction order *n* were obtained respectively according to the slope and intercept of the fitting line of $∆\mathrm{lg}\left(da/dT\right)/∆lg(1-\alpha )$ for ∆(1/T)/∆lg(1−α).

## 3. Results and Discussion

### 3.1. Vulcanization Characteristics

The vulcanization characteristics curve of EPDM is shown in Figure 1. The formulation of a semi-effective vulcanization system was used in this article (accelerator to sulfur dosage ratio of 1, which is between CV and EV). Both polysulphide bonds and a moderate amount of single and double sulfur crosslinked bonds existed in the resulting EPDM vulcanized rubber. The resulting EPDM vulcanized rubber has both polysulphide bonds and a moderate amount of single and double sulfur crosslinked bonds. This combines the characteristics of both CV and EV vulcanization systems to improve the thermal and oxygen aging and fatigue resistance of the vulcanized rubber. The change in torque during rubber vulcanization is proportional to the change in crosslinking density, and the rate of change in torque can be used to characterize the rate of vulcanization. The equation for the rate of vulcanization can be expressed as in Equation (15).
(15)$$V=\frac{dln\left(M_{H}-M_{t}\right)}{dt}=K{\left(M_{H}-M_{t}\right)}^{n}$$
where *M_H_* is the maximum torque (Nm); *M_t_* (Nm) is the torque at vulcanization time *t* (min); *K* is the rate constant; *n* is the number of reaction stages.

According to the reaction model for the coking phase derived by Coran [27], the reaction equation for the vulcanization phase can be expressed as in Equation (16).
(16)$$V_{ut}=V_{u\infty }[1-e^{-K_{2}\left(t-t_{i}\right)}]$$

In addition, the reaction during the scorch period is not a primary reaction, but rather the true primary reaction begins when the rate of change of the torque reaches a maximum, and the sulfation kinetics can be expressed as
(17)$$ln\left(M_{H}-M_{t}\right)=lnA-K(t-t_{dis})$$

In this case, *t_dis_*, the moment when the rate of change of torque reaches its maximum, is plotted against (*t* − *t_dis_*) by *ln*(*M_H_ − M_t_*) (Figure 2). *K* and *lnA* can be obtained from the slope and intercept, respectively.

As the sulfidation process after coking was usually divided into two stages. After linear fitting the data, it was found that the result was a smooth straight line, which was consistent with a first-order kinetic reaction. Therefore, this sulfidation process can be divided into two first-order kinetic reactions.

EPDM vulcanization rubber characteristics data are shown in Table 2. Due to the addition of antioxidant, *T*_10_, *T_dis_* were shortened, indicating that the start reaction time was earlier. The reaction difficulty was lower than Sample 1 with increasing vulcanization rate K, CRI value, and vulcanization speed. However, the crosslink density was lower than in Sample 1.

### 3.2. Physical Properties and Crosslinking Density

The physical properties and crosslinking density changes of EPDM vulcanizates after aging are shown in Figure 3. It can be seen that the tensile strength, hardness, and crosslinking density of two EPDM vulcanizates gradually increase with increasing aging time. However, the elongation at break decreased with increasing aging time. Basically, EPDM vulcanizates would continue to be crosslinked at 120 °C, the conformation transition of the polymer chain was constrained, and the flexibility was weakened, which led to the increase in tensile strength and surface hardness [4]. The polarity would increase, and the flexibility would decrease due to the oxidative degradation of the carbon–carbon backbone during the thermal-oxidative aging of EPDM vulcanizates, leading to lower elongation at break.

The elongation at break of Sample #2 was higher than that of Sample #1, while the tensile strength, crosslinking density, and hardness were slightly lower than that of Sample #1. It indicated that antioxidants had an effect on the vulcanization characteristics, which can inhibit the generation of crosslinking network structure and a larger free volume of the macromolecular chain. The degradation reaction and crosslinking reaction of EPDM vulcanizates were competitive reactions during thermal-oxidative aging. Antioxidants increase the imperfection degree of crosslinking structure. At the same time, chain breaking and oxidation reactions play a leading role in increasing thermal oxygen aging time. Therefore, both the crosslinking and degradation are quick as the crosslinking density rapidly increases in 72 h, the hardness increases, and flexibility decreases rapidly. After 72 h, both crosslinking and degradation speed are reduced, and the hardness and flexibility change slowly, which is caused by the disappearance of free radicals.

### 3.3. Aging Coefficient and Retention

The aging coefficient (*AC*) was obtained from the relationship between physical and mechanical properties (e.g., tensile strength, elongation) before and after thermal and oxygen aging. It was usually a key indicator to evaluate the overall performance of rubber aging. The changes in the aging coefficient and tearing elongation retention at the break of EPDM vulcanizates with aging time are shown in Figure 4. The aging coefficient and elongation retention rate of Sample #2 were larger than those of Sample #1. It meant that the retention rate of the macro physical properties of EPDM vulcanized rubber with thermal and oxygen aging was improved because of the addition of antioxidants.

### 3.4. Attenuated Total Reflection Fourier Transform Infrared Spectrometer (ATR-FTIR) Analysis

In this paper, the structure of EPDM vulcanizates was analyzed by the ATR-FTIR method after aging. The change of the infrared spectrum of EPDM vulcanizates is shown in Figure 5 with aging time. The same change rule of infrared absorption peaks can be observed at different wavelengths for two samples.

The peak at 3300 cm^−1^ was the stretching vibration peak of the hydroxyl group, which was attributed to the extended oxidation group of -OH (alcohol, hydroperoxide, and carboxylic acid) [28]. As a result of thermal-oxidative degradation, which content increased with increasing aging time. For pure EPDM, the peaks between 2800 cm^−1^ and 3000 cm^−1^ were related to CH_3_, CH_2_, and CH [29]. The peak at 2850 cm^−1^ and 2920 cm^−1^ belonged to the symmetric stretching vibration and asymmetric stretching vibration of the (-CH_2_-) group, respectively [30], which decreased with aging time. The peak at 2960 cm^−1^ reflects the asymmetric stretching vibration of methyl [31,32,33,34]. Its content increased with the extension of the aging time, indicating a chain-rupturing reaction of the EPDM structure.

The peak at 1720 cm^−1^ was the characteristic peak of carbonyl, in which the peak area was a measure of the oxidative degradation degree of the polymer [35,36,37]. As carbonyl was the main product of oxidative degradation of EPDM vulcanizates, the carbonyl index can be determined, namely, the value of carbonyl sulfide line strength at 1720 cm^−1^ divided by methylene line strength at 1432 cm^−1^ [38]. Carbonyl content and carbonyl index were shown in Table 3. The upward trend of carbonyl content and carbonyl index with increasing aging time implied that EPDM vulcanizates were gradually oxidized and degraded.

The carbonyl content and carbonyl index of Sample #2 with the addition of antioxidants were smaller than those of Sample #1, indicating that the addition of RD and 4010NA antioxidants can reduce the oxidation reaction on the surface of EPDM vulcanized rubber to a certain extent and improve the resistance of heat and oxygen aging.

In the absorption region of 1500–800 cm^−1^, the peak at 1432 cm^−1^ was the -CH_2_- shear vibration mode of the propylene unit of EPDM and CH band loss due to oxidative degradation [39]. The peak at 1245 cm^−1^ was caused by macromolecular skeleton vibration, and the content decreased with increasing aging time also implies that thermal oxygen aging would lead to the fracture of the macromolecular skeleton.

The peaks at 872 cm^−1^ and 802 cm^−1^ were the deformation vibration of the typical unsaturated zone C=C-H of the diene part, and 973 cm^−1^ was the shear deformation vibration of C-H in the third monomer with double bond C=C-H. These content increased first and then decreased with the increasing aging time, which was due to crosslinking firstly, and then breakage loss of double bonds caused by the thermal-oxidative aging degradation [40,41]. The carbonyl and ester groups were formed, and the unsaturation of the third monomer decreased rapidly, which indicated that chain branching [42] and crosslinking reactions [43] occur simultaneously during the aging process of EPDM vulcanizates. Therefore, thermal-oxidation aging of EPDM vulcanizates can be seen as a process of breaking the main chain and oxidation.

### 3.5. Surface Property

The contact angle is an important parameter to measure the wetting performance of the liquid on the surface of the material and can express the change of polarity, hydrophilicity, and surface energy of the material surface.

Variations in the contact angle of EPDM vulcanized rubber with various aging times are shown in Figure 6, with the specific data in Table 4. The gradual decrease in contact angle for two types of EPDM vulcanized rubber can be seen with the increase in aging time.

The reason was that the main chain of EPDM vulcanized rubber was broken and oxidized with increasing oxygen-containing groups (such as carbonyl and carboxyl groups on the surface). As a result, the number of carbon atoms in the main chain would decrease with increasing water solubility, indicating the hydrophilic energy gradually increased with decreasing contact angle. As the aging time is further prolonged, the aging reaction of ethylene propylene diene monomer would intensify, and the oxidative degradation of the main chain of EPDM would further intensify, leading to an increase in the carbonyl index, resulting in the decreased contact angle.

The contact angle of Sample #2 with the addition of antioxidants was larger than Sample 1. It was probably due to the fact that the addition of the antioxidant affects the crosslinking reaction and network of the EPDM vulcanized rubber while reducing the oxidation reaction during the vulcanization process. Further, the oxidative degradation of the EPDM vulcanized rubber surface was more intense as the aging time increased. It was suggested that the addition of antioxidants RD and 4010-NA could significantly reduce the generation of oxidation groups on the surface with an increasing carbonyl index.

### 3.6. Thermal Degradation Behavior and Kinetic

The thermal stability of materials can be analyzed by TGA, which can reveal the relationship between the mass of the sample and temperature or time under a certain flow rate atmosphere. The TGA test was the change of sample quality at the same heating rate and isothermal time. Different stages of thermal decomposition of the sample can be seen in the TG curve with the quantitative analysis and qualitative analysis of the composition change after aging. The DTG curve represents the weight-loss rate, which can be derived from the TG curve, and the two curves correspond. The decomposition stage, decomposition temperature, decomposition rate, thermal stability, and thermal decomposition activation energy of polymer materials can be obtained by thermogravimetric analysis. The TG/DTG curve of EPDM vulcanizate after aging is shown in Figure 7.

According to the TG/DTG curve in Figure 7, the thermal weightlessness of EPDM vulcanizates can be divided into three stages. The specific data are listed in Table 5. It can be seen that the changing trend of the initial weight-loss temperature in the first stage gradually reduced with increasing aging time. However, no change can be seen with time in the second and third stages. The initial weight-loss temperature in the first stage also represents that its thermal stability gradually decreases. The weight-loss rate of EPDM vulcanizates in the first stage was gradually reduced with increasing aging time. The major losses were highly volatile substances (plasticizers, accelerators, water vapor, etc.), whose aging was accompanied by quality degradation in the first stage at 120 °C. In the second stage, medium volatile substances (EPDM rubber) lost away largely. Some scholars [4] tracked the degradation and pyrolysis process of EPDM by gas chromatography to confirm that the main products were CO_2_, H_2_O, CO, CH_4_, and C_2_H_6_. These combustible substances (carbon black, etc.) were burned slowly in the third stage. The remaining materials were some mineral additives, such as ZnO.

The peak of the first weight-loss temperature increased with increasing aging time, while the peak of weight-loss temperature in the second and third stages did not change significantly. It was attributed to the loss of plasticizers, crosslinking, and degradation (the competitive reaction), which were caused by the decomposition of the carbon main chain, affecting the free volume of the material during the thermal-oxidative aging process of EPDM vulcanizates. The degradation was attributed to the oxidative degradation of the main chain of crosslinked EPDM in the second stage [44].

In this paper, the temperature interval 450–480 °C was chosen to calculate the heat drop activation energy. This interval was mainly attributed to the EPDM main chain structure, where the weight-loss reaction was relatively intense. The activation energy E and the number of reaction steps n can be obtained from the slope and intercept of the straight line by the Freeman–Carroll method for graphing, respectively (Figure 8). The data of the thermal analysis of the TG/DTG curve of EPDM vulcanization rubber aging can be seen in Table 6.

It can be seen in these results that the degradation process of EPDM was complex, and which degradation reaction was not a simple stepwise reaction but a random degradation process [45]. The degradation of EPDM follows the Avrami-Erofeev two-dimensional nucleation model or random chain-breaking mechanism [46]. The activation energy E of thermal degradation of the EPDM vulcanized rubber backbone structure tends to decrease with increasing thermal and oxygen aging time. This is due to the dual effect of chain breakage and crosslinking occurring simultaneously during thermal oxygen aging with the continuous crosslinking reaction of EPDM vulcanized rubber at 120 °C under thermal oxygen [39]. The increase in crosslinking density leads to a downward trend in the activation energy of thermal degradation.

However, as the aging time increases, the oxidative degradation reaction of the EPDM backbone dominates, and the activation energy continuously decreases [44,47].

It can be seen in the comparison of the results from the #2 (with antioxidant) and #1 (without antioxidant) samples that the EPDM vulcanization starting weight-loss temperature and the first-stage weight-loss peak temperature was risen because of the addition of antioxidants.

At the same time, the number of reaction stages was reduced, indicating that the thermal degradation reaction and the oxidation effect were weakened due to the existence of antioxidants in Sample 2. This was generally consistent with the macroscopic physical and mechanical properties. The activation energy of thermal degradation of Sample 2 was smaller than that of Sample 1. It was attributed to the competition between degradation and crosslinking of the EPDM backbone. The antioxidant enhanced the crosslinking rate without help to the complete crosslinking network of the EPDM curing rubber, reducing the stability of the carbon–carbon backbone. Herein, the degradation reaction played a major role in reducing the thermal degradation activation energy.

## 4. Conclusions

The structure and properties of EPDM vulcanizates with two kinds of semi-efficient vulcanization systems before and after thermal-oxidative aging were investigated in detail, including vulcanization characteristics, macroscopic physical properties, surface properties, thermogravimetric behavior and kinetics, and so on.

The vulcanization characteristics showed that the vulcanization process of EPDM vulcanizates consisted of two first-order kinetic reaction stages. Antioxidants can promote the crosslinking reaction and improve the crosslinking speed but cannot form perfect crosslinking network structures. With increasing aging time, EPDM vulcanizates can continue to crosslink so that the conformational transformation of the polymer chain is constrained and the flexibility is weakened. Finally, the macroscopic physical properties (such as strength, elongation, hardness, etc.) were affected. The antioxidant can significantly improve the aging coefficient and macroscopic physical retention of EPDM vulcanizates, which can lower (reduce or decrease is OK) crosslinking density to produce larger free volume, promoting the retention of elongation and hardness.

The same change of the infrared absorption peaks of two EPDM vulcanizates at different wavelengths was shown in FTIR, which implied that the oxidation and degradation of the surface intensify as the aging time increases. Moreover, the carbonyl content and carbonyl index gradually increase with increasing aging time. This phenomenon is demonstrated macroscopically as an increase in contact angle. The degree of oxidation and degradation of the surface of EPDM vulcanizates can be reduced by antioxidants. Moreover, the thermal oxygen aging properties of the surface can be improved due to the addition of antioxidants.

Three stages of the thermal decomposition of two EPDM vulcanizates can be seen in TGA results. Sample #2, with the addition of the high boiling-point antioxidant, effectively increased the initial weight-loss temperature and the peak of the first weight-loss temperature in the first degradation stage. It can be seen in the kinetic study of the backbone of EPDM vulcanizates under the temperature range of 450–480 °C that the addition of antioxidants can reduce the reaction order and promote the formation of incomplete crosslinked network structure, which intensifies the degradation reaction and reduces the thermal degradation activation energy of the backbone.

## Data Availability

Available data are presented in the manuscript.

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
