# Peer review of "Performance of Thermal-Oxidative Aging on the Structure and Properties of Ethylene Propylene Diene Monomer (EPDM) Vulcanizates"

_polymers, 2023, doi:10.3390/polym15102329_

Round 1

Reviewer 1 Report

1.     The units in Table 1 are wt%. I think there can be replaced by “phr”, which is more suitable. Because the total content of all the materials in Table 1 is more than 100.

2.     In line 199-124, the unites of the parameters should be marked。

3.     Figure 3 and Table 3, Figure 4 and Table 4 contain the same data. Just one is OK.

4.     In line 247-248, the authors said “while aging produces more silver lines and increases the number of weak points leading to lower elongation at break. “ if the number of weak points increased as aging time increased, why the tensile strength of the sample increased? In my opinion, the tensile strength will be decreased as the weak points increased.

5.     In line 255-256, the authors said “Antioxidants increase the imperfection degree of cross-linking structure, at the same time, chain breaking and oxidation reaction play a leading role with increasing thermal-oxygen aging time.” Antioxidants can capture free radicals, and after the disappearance of free radicals, both crosslinking and degradation will be reduced. Therefore, the author needs to further elaborate.

6.     Figure 6 has not been mentioned in the manuscript.

  Please check the format, grammar, and spelling of the article. There are some errors.

Reviewer 2 Report

Authors deserve to have their work accepted for publication. But there are a few things to note.

1. In the introduction, there is little to no information about the antioxidants used for the EPDM polymer. At the same time, there are quite a lot of such works. I ask the authors to describe at least the main types of antioxidants used.

2. There are many inaccuracies in the text of the article: chapters 2,3,7 and 2,3,8 have the same title, figure 6 has no description and is placed above the Surface property chapter.

3. In the Fourier Transform Infrared Spectrometer (FTIR) Analysis chapter, replace Infrared spectrum (spectra) with ATR-FTIR spectrum (spectra). Because IR spectra recorded using ATR equipment are different from IR spectra. Also, there is no description for ATR accessory.

Round 2

Reviewer 1 Report

The author has carefully revised this manuscript.  I suggest accepting it.